# A Marine Alkaloid, Ascomylactam A, Suppresses Lung Tumorigenesis via Inducing Cell Cycle G1/S Arrest through ROS/Akt/Rb Pathway

**DOI:** 10.3390/md18100494

**Published:** 2020-09-27

**Authors:** Lan Wang, Yun Huang, Cui-hong Huang, Jian-chen Yu, Ying-chun Zheng, Yan Chen, Zhi-gang She, Jie Yuan

**Affiliations:** 1Department of Biochemistry, Zhongshan School of Medicine, Sun Yat-sen University, Guangzhou 510080, China; wlwlan3435@163.com (L.W.); wyhl2014@163.com (Y.H.); yujchen@mail2.sysu.edu.cn (J.-c.Y.); 2Department of Pathogen Biology and Immunology, School of Life Sciences and Biopharmaceutics, Guangdong Pharmaceutical University, Guangzhou 510006, China; zyc1502667588@163.com; 3School of Basic Medical Sciences, Southern Medical University, Guangzhou 510515, China; 4School of Medicine & Health Care, Shunde Polytechnic, Shunde 528333, China; hch@sdpt.edu.cn; 5Key Laboratory of Tropical Disease Control (Sun Yat-sen University), Ministry of Education, Guangzhou 510080, China; 6School of Chemistry, Sun Yat-sen University, Guangzhou 510275, China; chenyan27@mail2.sysu.edu.cn

**Keywords:** ascomylactam A, lung cancer, cell proliferation, cell cycle arrest, ROS, Akt, Rb

## Abstract

Ascomylactam A was reported for the first time as a new 13-membered-ring macrocyclic alkaloid in 2019 from the mangrove endophytic fungus *Didymella* sp. CYSK-4 from the South China Sea. The aim of our study was to delineate the effects of ascomylactam A (AsA) on lung cancer cells and explore the antitumor molecular mechanisms underlying of AsA. In vitro, AsA markedly inhibited the cell proliferation with half-maximal inhibitory concentration (IC_50_) values from 4 to 8 μM on six lung cancer cell lines, respectively. In vivo, AsA suppressed the tumor growth of A549, NCI-H460 and NCI-H1975 xenografts significantly in mice. Furthermore, by analyses of the soft agar colony formation, 5-ethynyl-20-deoxyuridine (EdU) assay, reactive oxygen species (ROS) imaging, flow cytometry and Western blotting, AsA demonstrated the ability to induce cell cycle arrest in G1 and G1/S phases by increasing ROS generation and decreasing of Akt activity. Conversely, ROS inhibitors and overexpression of Akt could decrease cell growth inhibition and cell cycle arrest induced by AsA. Therefore, we believe that AsA blocks the cell cycle via an ROS-dependent Akt/Cyclin D1/Rb signaling pathway, which consequently leads to the observed antitumor effect both in vitro and in vivo. Our results suggest a novel leading compound for antitumor drug development.

## 1. Introduction

Lung cancer is the second most commonly diagnosed malignant tumor in men and women, and is one of the major causes of cancer deaths in the world. In 2017, there were an estimated 1.5 million new incident cases of tracheal, bronchus, and lung (TBL) cancer, with an approximated 1.3 million male deaths and 596,000 female deaths in 195 countries [1]. Furthermore, only 19% of all lung cancer patients survived for 5 years during the period 2009–2015 [2]. Approximately 85% of patients have a group of histological subtypes collectively known as non-small cell lung cancer (NSCLC), which mainly including lung adenocarcinoma (LUAD) and lung squamous cell carcinoma (LUSC) [3]. Over the past 20 years, although treatment has evolved from the empiric use of cytotoxic therapies to more effective targeted therapy, unfortunately, almost all cases eventually recur after a median of approximately 10 months from the onset of treatment, which becomes the greatest challenge for the therapy in lung cancer [4]. Accordingly, discovery and development of novel agents with low toxicity remains a therapeutic challenge in the treatment of NSCLC.

A great number of antitumor compounds are developed from natural products, such as doxorubicin, rapamycin, and mithramycin [5,6]. Moreover, natural compounds, such as 15d-PGJ2, luteoloside and isoacteoside, have been reported to be able to induce cell cycle arrest through the reactive oxygen species (ROS)-mediated inactivation of Akt [7,8,9]. Cancer cells are more susceptible to oxidative stress than normal cells because of genetic alterations and abnormal growth. However, as excessive levels of ROS stress can also be toxic to the cells, cancer cells with increased oxidative stress are likely to be more vulnerable to damage by further ROS insults induced by exogenous agents [10,11]. Therefore, manipulating ROS levels by redox modulation is a way to selectively kill cancer cells without causing significant toxicity to normal cells.

Secondary metabolites from mangrove endophytic fungi have proven to be an interesting and significant source for drug discovery, which always have structural novelty and biological activity [12]. Ascomylactam A (AsA) is a 13-membered-ring macrocyclic alkaloid isolated from the mangrove endophytic fungus *Didymella* sp. CYSK-4, which was obtained from Shankou Mangrove Nature Reserve in Guangxi Province, China. AsA, as a decahydrofluorene analogue with a tetracyclic skeleton fused with a 13-membered macrocyclic moiety, is relatively rare in the decahydrofluorene class. We recently demonstrated that AsA can effectively suppress the growth of cell lines derived from a variety of human tissues, including MDA-MB-435, HepG2, HCT116, and NCI-H460 [13]. In the present study, we found that AsA could inhibit the proliferation of lung cancer cells and suppress tumor cell growth in xenograft mouse models without obvious toxicity. Further studies revealed that AsA treatment resulted in intracellular ROS production, regulation of the Akt/Cyclin D1/Rb pathway, and cell cycle G1/S phase arrest, which might be the underlying mechanism of the AsA anticancer activity in vitro and in vivo.

## 2. Results

### 2.1. AsA Inhibits the Proliferation of Lung Cancer Cells In Vitro

To investigate the effect of AsA (Figure 1a) on lung cancer cells, we firstly determined the cytotoxicity by 3-(4,5-dimethylthiazol-2-yl)-2,5-diphenyl tetrazolium bromide (MTT) assay. After 48 h treatment with AsA, the growth of lung cancer cells was markedly inhibited by AsA in a concentration-dependent manner and the half-maximal inhibitory concentration (IC_50_) values of AsA ranged from 4 to 8 μM for six lung cancer cell lines, respectively (Figure 1b). Morphological changes were observed by phase-contrast microscope, which were induced by the increasing the concentration of AsA for 4 h in A549, NCI-H460 and NCI-H1975 cells (Figure 1c). To further confirm the inhibition of cell proliferation by AsA in lung cancer cells, the colony formation assay and soft agar colony formation assay were conducted on A549, NCI-H460 and NCI-H1975 cells. As shown in Figure 1d, the clone formation abilities of the cells were clearly suppressed by incubation of AsA. In addition, the anchorage-independent capacity for cell growth of the cells was also reduced by the treatment of AsA in a dose-dependent manner (Figure 1e).

### 2.2. AsA Suppresses NSCLC Cells Growth In Vivo

To evaluate the anticancer properties of AsA in vivo, we implanted xenografts of A549, NCI-H460 and NCI-H1975 cells into nude mice. When the xenograft tumors grew to 80–100 mm^3^ in size, the mice were randomly assigned into four groups and treated with vehicle, DDP (cisplatin, 5 mg·kg^−1^), and AsA (3 mg·kg^−1^, 6 mg·kg^−1^, or 5 mg·kg^−1^, 10 mg·kg^−1^) once every three days. The results demonstrated that AsA treatment strongly inhibited tumor growth in vivo (Figure 2a). In parallel, at the end of the experiment, the excised tumors in AsA treatment groups had a lower weight and smaller size than tumors in control groups (Figure 2b,c). The weight gain curves of the mice fit closely together, with the exception of the DDP 5 mg·kg^−1^ group in xenograft mice of A549 (Figure 2d). Collectively, our findings demonstrated a potent in vivo therapeutic efficacy of AsA, and the effective dose of AsA (3–10 mg·kg^−1^ via i.p.) displayed no detectable adverse effect as the mice showed no significant weight loss compared to the control groups, which indicates that this compound might have the features for further clinical drug development.

### 2.3. AsA Induces G1/S Cell Cycle Arrest in Lung Cancer Cells

To explore the mechanisms underlying how AsA inhibited the growth of lung cancer cells, first we used the EdU assay to measure the effect of AsA on DNA synthesis in treated cells. Results showed that AsA dramatically inhibited DNA synthesis in A549, NCI-H460 and NCI-H1975 cell lines in a dosage-dependent manner (Figure 3a–d). Additionally, the analysis of the cell cycle distribution by flow cytometry, with the treatment of AsA for 12 h against A549 cells, resulted in an increased percentage of the cell population in the G0/G1 phase (from 52.69% in untreated cells to 53.78, 63.79, and 81.94% in treatment cells with 1, 2.5 and 5 μM of AsA, respectively) and a decreased percentage in S phases (from 32.91% in untreated cells to 33.14, 27.84, and 12.03% under the same treatments) and G2/M phases (from 14.39% in untreated cells to 13.08, 8.37, and 6.03% under the same treatments) (Figure 3e). Similar experiments were performed in NCI-H460 and NCI-H1975 cells generating similar results (Figure 3e). The results suggested that AsA effectively induced G1/S cell cycle arrest in a dose-dependent manner.

Based on the results of cell cycle distribution, we hypothesized that AsA should affect the expression of cell cycle-regulatory genes, such as cyclins and CDKs, which are essential for cell cycle progression from G1 to S phase. Indeed, we found that AsA remarkably reduced the protein levels of CDK4, CDK6 and Cyclin D1 which were involved in the G1 phase checkpoint, in a dose- (Figure 3f) and time-dependent manner (Figure 3g). Furthermore, we examined the levels and phosphorylation status of Retinoblastoma protein (Rb) in these cells, which is inactivated by CDK-dependent phosphorylation or by proteolytic degradation. AsA treatment strongly decreased levels of phosphorylation of Rb at serine residue 780 (p-Rb^Ser−780^) in A549, NCI-H460 and NCI-H1975 but the changes in protein expression of total Rb were less pronounced (Figure 3f,g). Taken together, these data indicated that the suppression on proliferation of lung cells by AsA is correlated with the G1/S arrest induced by the effect of AsA on the cell cycle regulation proteins.

### 2.4. AsA Blocks the Cell Cycle via Upregulating ROS

Accumulating studies have reported that various natural products exhibited powerful antitumor effects by the generation of ROS with their pro-oxidative activities [14,15,16]. In our investigation, the intracellular ROS production in the lung cancer cells was measured after treatment of AsA. As illustrated in Figure 4, AsA induced ROS production in A549, NCI-H460 and NCI-H1975 cells in a dose-dependent manner. To examine whether the enhanced ROS generation plays a role in AsA-induced cell cycle arrest, the growth and cell cycle in lung cancer cells were detected by proliferation assays and flow cytometry, respectively. The results showed that the inhibition induced by AsA on cancer cell growth was diminished by ROS scavengers, such as glutathione (GSH), catalase and superoxide dismutase (SOD), except for N-acetyl-L-cysteine (NAC) (Figure 5a). Additionally, ROS generation could be markedly reversed, when the cells were co-treated with catalase, one of the ROS inhibitors, which confirmed that AsA resulted in intracellular ROS accumulation (Figure 5b,c). Moreover, the suppression of expression of Cyclin D1 and p-Rb^Ser−780^ was eliminated by pretreatment with catalase for 2 h before AsA treatment (Figure 5d). Consistently, the cell cycle arrest was prevented under the co-treatment with AsA and catalase in A549, NCI-H460 and NCI-H1975 cells (Figure 5e). These results suggest that intracellular ROS activation is an essential event in the induction of cell cycle arrest by the AsA treatment.

### 2.5. AsA Induces the ROS-Dependent Inactivation of PI3K/Akt Pathway in Lung Cancer Cells

Abnormal activation of the PI3K/Akt signaling pathway has recently been shown to be involved in the pathogenesis of multiple human tumors, including lung cancer [17]. Indeed, we found a significantly decline in the expression of phosphate of Akt (p-Akt) in the AsA-treatment cells (Figure 6a). Furthermore, the elimination of ROS by catalase resulted in a reversion of the decrease in the p-Akt levels induced by the presence of AsA (Figure 6b), which confirmed that ROS were involved in the effects of AsA-dependent inactivation of Akt. However, the inhibition of Akt by MK2206 did not affect ROS activation (Figure 6c). These data suggest that Akt was a downstream molecule of ROS. Moreover, compared with parental cells, overexpression of Akt reversed the cell cycle G1/S arrest induced by AsA in A549-myr-Akt cells (Figure 6d). Moreover, in the A549-myr-Akt cells, the downregulation of the proteins involved in G1/S phase transition, such as p-Akt, CDK4, CDK6, Cyclin D1 and p-Rb^Ser−811^, were eliminated by overexpression of Akt (Figure 6e). Taken together, these data suggest that ROS acted upstream of the Akt pathway to regulate the cell cycle and the ROS/Akt/Rb pathway-mediated AsA-induced cell cycle G1/S arrest in lung cancer cells. 

Altogether, these results indicate that AsA treatment could effectively inhibit lung cancer cell growth in vitro and in vivo, and the mechanistic study showed that AsA induced G1/S cell cycle arrest by upregulating ROS and decreasing p-Akt (Figure 6f).

## 3. Discussion

Natural products (secondary metabolites), especially marine natural products, have been a rich source of novel drug leads [18,19]. Several natural products with the similar structures of AsA have been reported presenting a variety of biological activities. Pyrrospirones A and B, as well as pyrrocidine A, were apoptosis inducers in HL-60 cells [20,21]. Embellicines A and B were cytostatic and inhibited NF-κB transcriptional activity [22]. Besides, phomapyrrolidone A was a p97 ATPase inhibitor [23]. In this study, we first reported that AsA suppressed tumor growth in xenograft mouse models and significantly inhibited the proliferation by inducing cell cycle G1/S phase arrest in lung cancer cells. These studies suggest that AsA and its structural analogues have good antitumor drug activity with different modes of action, and the core skeleton may be the characteristic structure to develop antitumor lead compounds.

In tumor cells, an uncontrolled cell cycle is a significant indicator and the restriction function of the G1-S phase check point is reduced in cancer cells, which often leads to uncontrolled cell proliferation [24]. The data revealed that AsA decreased cyclin D, CDK4, p-RB and induced significant G0/G1-phase accumulation compared with the control in NSCLC cells. These findings provided evidence that AsA inhibited the growth of lung cancer cells through the G1/S phase arrest. Furthermore, when the AsA concentration was raised to 15 μM, we also detected an increase in activating cleaved bands of caspase-3 and PARP, which are the critical markers of apoptosis (Appendix A). These results indicate that, besides the cell cycle effects, AsA probably also affects cell viability. It can be speculated that the mechanism of anti-tumor activity of AsA in vivo may not be limited to the cell cycle arrest. In this context, further effort in discovering the molecular mechanism of antitumor of AsA is needed in future investigations.

ROS play critical roles in the regulation of proliferation, apoptosis, and cellular transformation [25,26,27]. It has been demonstrated that ROS are implicated in diverse diseases, including cancers. Compared with their normal counterparts, cancer cells display inherently elevated ROS levels. Increased ROS levels are closely related to cancer initiation, metastasis, and drug resistance [28,29]. Although moderate increases in ROS levels can be beneficial to cancer cells, excessive amounts of ROS can cause cell death [30,31]. ROS-mediated cell cycle arrest has also been identified as an important mechanism in some antitumor agents, such as cardamonin and furanodienone [32,33]. In this study, we found that generation of ROS is intimately involved in cell cycle arrest induced by AsA and further investigation revealed that the cancer cell growth inhibition, ROS generation, and the cell cycle arrest could be markedly reversed by ROS scavengers, such as GSH, catalase and SOD. These results demonstrate that the cell cycle arrest was the downstream effect of ROS generation in NSCLC cells in response to AsA. However, we also recognize the difference in the effects of different ROS scavengers, which indicates further investigation should be carried out to realize the mechanism of ROS production induced by AsA.

Akt is a key proto-oncogene serine/threonine kinase termed protein kinase B (PKB). Key roles for this protein kinase in cellular processes are now well established, such as cell proliferation, glucose metabolism, apoptosis, transcription and cell migration [34]. In this study, AsA exposure resulted in diminished p-Akt. In addition, high expression of Akt could impair the cell cycle arrest induced by AsA. To further determine the relationship between p-Akt decrease and ROS generation in AsA-induced cell cycle arrest, catalase (an ROS scavenger) and MK2206 (an Akt inhibitor) were used to treat NSCLC cells. P-Akt levels were decreased after AsA treatment, and this decrease was reversed by catalase. However, AsA-induced ROS generation was not inhibited by MK2206 in NSCLC cells. These results revealed that ROS is upstream of the PI3K/Akt signaling pathway, while Akt does not affect the ROS accumulation.

In conclusion, our results demonstrated that AsA effectively inhibited the cell growth of lung cancer cells and that AsA could induce G1/S arrest and modulate the activity of Akt/Cyclin D1 pathway in lung cancer cells through an ROS-dependent mechanism. This is the first report that clearly characterizes the antitumor properties of AsA and identifies its mechanism in a tumor model. Our results on the activity and mechanism of AsA are interesting for future investigations, which also help to understand similar kinds of compounds and provide theoretical guidance to explore other novel clinical drugs.

## 4. Materials and Methods

### 4.1. Preparation of AsA

AsA was prepared and purified from mangrove endophytic fungus Ascomycota sp. CYSK-4 as previously reported, and its structure was identified by interpretation of spectral data (MS, 1H NMR, 13C NMR, 2D NMR) and X-ray single crystal diffractive technique) [13]. The compound was dissolved in 99.9% dimethyl sulfoxide (DMSO) at a concentration of 10 mM as stock solution and diluted according to experimental requirements when used.

### 4.2. Cell Culture

Human lung cancer cell lines A549, NCI-H460, NCI-H1299, NCI-H1975 and 95-D were obtained from the cell bank of the Shanghai Institutes of Biological Sciences (Shanghai, China) or from Fu Erbo Biotechnology Co., Ltd. (Guangzhou, China). NCI-H226 (CRL-5826) were purchased from the American type culture collection (ATCC, Manassas, VA, USA). Cells were cultured in Dulbecco’s modified Eagle’s medium (DMEM) (Invitrogen, Carlsbad, CA, USA) supplemented with 5% fetal bovine serum (Hyclone, Logan, UT, USA), 2 mM L-glutamine, 100 mg·mL^−1^ streptomycin and 100 units·mL^−1^ penicillin (Invitrogen, Carlsbad, CA, USA). The cultures were maintained at 37 °C in a humidified atmosphere of 5% CO_2_.

### 4.3. Cell Viability Assay

Cell viability was determined by 3-(4,5-dimethylthiazol-2-yl)-2,5-diphenyl tetrazolium bromide (MTT) reagent (Genview, Houston, TX, USA) assay as previously described [35]. Briefly, 1 × 10^4^ cells per well were seeded into 96-well plates and incubated for 24 h, followed by exposure to various concentrations of AsA for 48 h. Cell growth inhibition was determined by MTT reduction assay and the half maximal inhibitory concentration (IC_50_) was defined as the compound concentration required to inhibit the cell growth by 50% compared with the control assay. The assay was performed in triplicates in three independent experiments.

### 4.4. Clonogenic Assay

Tumor cells were plated in 6-well plates at a density of 1000 cells/well. After cells were attached overnight, AsA at indicated concentrations was added to plates. DMSO served as the control vehicle. Medium with AsA was replaced every 3 days for a period of 7–14 days. The cells were then fixed with methanol and stained with 0.1% crystal violet. Colonies were counted by Image J software (NIH, Bethesda, MD, USA).

### 4.5. Anchorage-Independent Soft Agar Assay

Cells were plated at a density of 2 × 10^4^ cells/well in 6-well plates with 2 mL bottom agar (0.66%) and 2 mL top agar (0.33%) in normal growth medium. Cells were fed every 3 days by addition of new layer of top agar, which contained several concentrations of AsA. At 14 days, cells were stained and imaged with an inverted microscope at 20 × magnification (Axio Observer ZI, Carl Zeiss AG, Oberkochen, Germany) and the number of colonies over 100 μm were counted.

### 4.6. 5-Ethynyl-20-Deoxyuridine Assay

Cells were plated at a density of 1×10^4^ cells/wells in 24-well plates. After cells were attached overnight, AsA at indicated concentrations was added to each well for 12 h. Then, the cells were incubated with 5-ethynyl-20-deoxyuridine (EdU, Ribobio) for 2 h, and processed according to the manufacturer’s instruction. After three washes with PBS, the cells were treated with 300 μL of 1× Apollo reaction cocktail for 30 min. Then, the DNA contents of the cells in each well were stained with 200 μL of Hoechst 33,342 (Sigma-Aldrich, Germany) (5 μg·mL^−1^) for 30 min and visualized under a fluorescence microscope at 20× magnification (Axio Observer ZI, Carl Zeiss AG, Oberkochen, Germany).

### 4.7. Cell Cycle Analysis

For measurement the effect of AsA on the cell cycle, cells were subjected to serum starvation for 24 h, and then cells were pretreated with or without AsA at final concentrations of 1.25, 2.5, and 5 μM or PD-0332991 at final concentrations of 0.05 μM for 12 h. Cells were harvested and cell-cycle distributions were analyzed on a FACSort flow cytometer as instructed by the manufacturer (Becton Dickinson, San Jose, CA, USA). We analyzed the cell cycle using ModFit software (Verity Software House, Topsham, ME, USA).

### 4.8. Western Blotting Analysis

The cells were seeded in 60 mm diameter plates at 1 × 10^6^ cells per well. After incubation for 24 h, the cells were treated with AsA at different concentrations or DMSO for 12 h. The cells were harvested and lysed after treatment then lysates were probed with primary antibodies and horseradish peroxidase (HRP) (1:2000; Bio-Rad Laboratories, Inc.) conjugated secondary antibodies. Primary antibodies including CDK4 (1:500; RRID: AB_397549) were purchased from BD Biosciences (San Jose, CA, USA). CDK6 (1:500; RRID: AB_10999714), Cyclin D1 (1:1000; RRID: AB_2750906) and GAPDH (1:2000; RRID: AB_11143050) were purchased from Abcam (Cambridge, UK). Phospho-Akt (Ser473) (1:500; RRID: AB_2315049), Akt (1:500; RRID: AB_2093915), phospho-Rb (Ser780) (1:500; RRID: AB_330015), phosphor-Rb (Ser807/811) (1:500; RRID: AB_331472) were purchased from Cell Signaling Technology (Beverly, MA, USA). Rb (1:500; RRID: AB_628209) was purchased from Santa Cruz Biotechnology (Santa Cruz, CA, USA). α-Tubulin (1:2,000; RRID: AB_477593) was obtained from Sigma Aldrich (St Louis, MO, USA).

### 4.9. Detection of Intracellular ROS

ROS levels were measured using a flow cytometer (Beckman Coulter, Miami, FL, USA). Following treatment, cells were harvested and washed with PBS and then suspended in serum-free DMEM containing 10 μM of 2′,7′-dichlorodihydrofluorescein diacetate (DCFH-DA) at 37 °C, 5% CO_2_ for 30 min. Cells were washed with PBS, and flow cytometry analysis was performed with an excitation wavelength of 488 nm and emission wavelength of 525 nm.

### 4.10. Chemicals and Fluorescent Probes for Studying ROS Generation

The fluorescent dyes were obtained from Molecular Probes (Eugene, Org, USA). With regards to loading conditions for the fluorescent probes, the nuclei were stained with Hoechst 33,342 and the stained cells were observed under a fluorescence inverted microscope using filter set for FITC, while the ROS probe DCFH-DA was used at 10 μM. Cells were incubated with the fluorescent probes at 37 °C, 5% CO_2_ for 30 min, and then rinsed with PBS saline solution. Subsequently, cells were mounted on a cell chamber, and visualized using an inverted fluorescence microscope and output imaging system (Zeiss, Oberkochen, Germany) at Ex/Em = 355/465 nm (Hoechst 33342) and Ex/Em = 488/525 nm (FITC).

### 4.11. In Vivo Assay

All animal care and experimental procedures were approved by the Institutional Animal Care and Use Committee of Sun Yat-sen University. Male BALB/c-nu mice (18–20 g) were purchased from the Beijing Vital River Laboratory Animal Technology Co. Ltd. (Beijing, China), and were housed in a specific pathogen-free (SPF) environment. To assess the effectiveness of AsA against lung cancer in vivo, we used three different lung cancer cell lines to build xenograft mouse models, including A549 cells (2.5 × 10^6^ cells in 0.1 mL per mouse), NCI-H460 cells (1 × 10^6^ cells in 0.1 mL per mouse) and NCI-H1975 cells (3 × 10^6^ cells in 0.1 mL per mouse). For each of the in vivo assays, cancer cells were inoculated subcutaneously in the right side of axillary. When tumor volume reached to 80–100 mm^3^, the mice were randomly divided into four experimental groups. The control group received 200 μL of vehicle (2%DMSO, 15%Tween 80, 83%physiological saline) i.p. per mouse every 3 days for 21 days. The two treatment groups received 200 μL of AsA i.p. at a doses of 3 mg·kg^−1^, 6 mg·kg^−1^, or 5 mg·kg^−1^, 10 mg·kg^−1^ of body weight, respectively, i.p. per mouse every 3 days for 21 days. The positive control group was given 200 μL of DDP at 5 mg·kg^−1^ of body weight i.p. every 3 days for 21 days. Tumors were measured every 3 days in double-blinded manner, and the tumor sizes were assessed by measuring perpendicular diameters with a digital caliper. The tumor volumes (mm^3^) were calculated using the following formula: volume = π·6^−1^ × (length) × (width)2. Data were presented as the mean SD of six mice in each group. At the end of the experiment, all the animals were humanely euthanized, and the xenograft tumors were dissected and weighed.

### 4.12. Statistical Analysis

All experiments were performed at least three times, and data were expressed as mean ± SD. Student’s t-test was used to compare the differences between two groups. We compared more than two groups with one-way ANOVA with Tukey’s post hoc test, the overall F test was significant (*p* < 0.05), and there was no significant variance in homogeneity. GraphPad Prism version 5.0 (GraphPad Software, San Diego, CA, USA) was employed to conduct the statistical analysis. All statistical tests were two-sided, and *P* < 0.05 was considered statistically significant. The data and statistical analyses comply with the recommendations on experimental design and analysis in pharmacology.

### 4.13. Materials

PD 0332991(CDK4/6 inhibitor; #HY-50767, CAS Number: 571190-30-2) was purchased from MedChemExpress LLC (Shanghai, China). DDP (cisplatin; #BP809, CAS Number: 15663-27-1), N-Acetyl-L-cysteine (NAC; #A7250, CAS Number: 616-91-1), 2′,7′-dichlorodihydrofluorescein diacetate (DCFH-DA; #287810, CAS Number: 4091-99-0) and Dimethyl sulfoxide (DMSO; #D2650, CAS Number: 67-68-5) were purchased from Sigma-Aldrich (St. Louis, MO, USA). Cell cycle assay kits (#KGA512) were purchased from KeyGEN BioTECH (Jiangsu, China). Superoxide Dismutase (SOD; # S8410, CAS Number: 9054-89-1) was purchased from Beijing Solarbio Science & Technology Co. (Beijing, China). Catalase (#MB3116, CAS Number: 9001-05-2) and Glutathione (GSH; # 571190-30-2, CAS Number: 70-18-8) were purchased from Dalian Melun Biotechnology Co. (Dalian, China). MK2206 (Akt inhibitor; # S1078, CAS Number: 1032350-13-2) was purchased from Selleck Chemicals (Shanghai, China). A549-myr-Akt cells were a kind gift from Dr Junchao Cai, Sun Yat-sen University Zhongshan School of Medicine [36].

## Figures and Tables

**Figure 1 marinedrugs-18-00494-f001:**
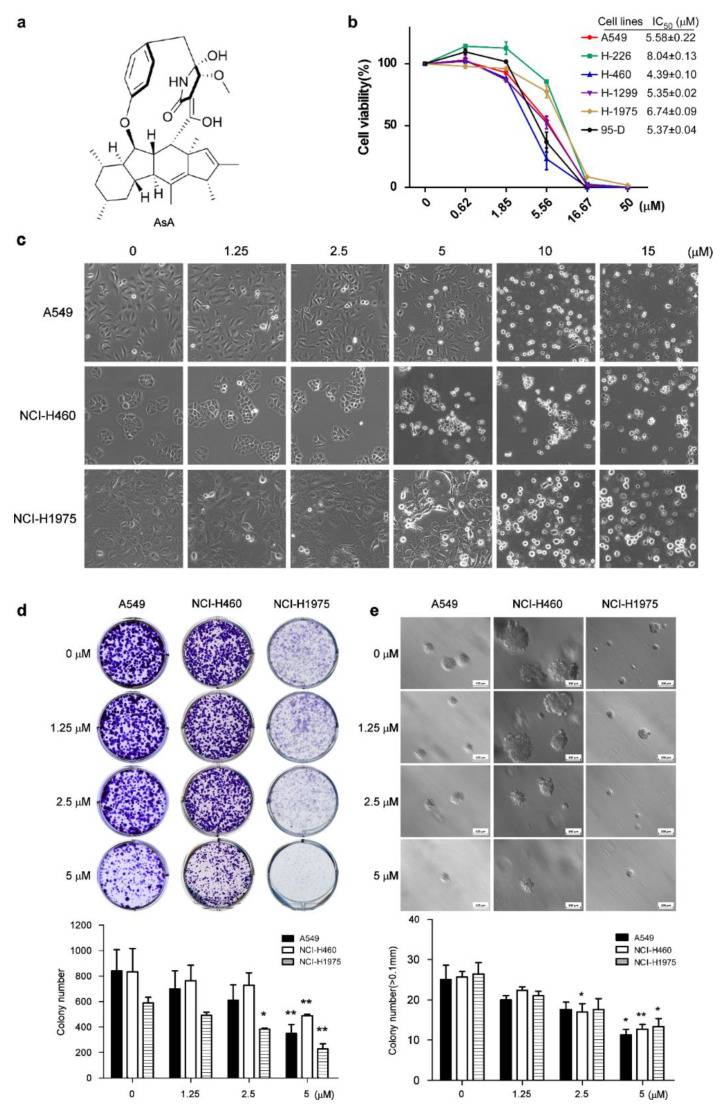
Ascomylactam A (AsA) significantly inhibits the proliferation of lung cancer cells. (**a**) Chemical structure of AsA. (**b**) Cell viability of a variety of lung cancer cells shown in the figure treated by AsA for 48 h detected by3-(4,5-dimethylthiazol-2-yl)-2,5-diphenyl tetrazolium bromide (MTT) assays. The bar shown represents the mean ± SD of samples from three wells. Data are representative of at least three independent experiments. (**c**) Morphological changes of A549, NCI-H460 and NIC-H1975 cells treated with AsA at indicated concentrations for 4 h observed by phase-contrast microscopy (magnification, 100×). The images shown here are representative of three independent experiments with similar results. (**d**) Clone formation efficiency of the cells treated by AsA. A549, NCI-H460 and NCI-H1975 cells were incubated with AsA at indicated concentrations in plates for 2 weeks. * *p* < 0.05, ** *p* < 0.01. (**e**) The anchorage-independent growth capacity measured by soft agar colony formation assay. A549, NCI-H460 and NCI-H1975 cells were incubated with AsA at indicated concentrations in soft agar plates for 3 weeks. The colonies were counted, and the data were plotted. * *p* < 0.05, ** *p* < 0.01. (**d**,**e**) Colony formation assay and soft agar assay data are mean ± SD and representative of ≥ 3 experimental replicates.

**Figure 2 marinedrugs-18-00494-f002:**
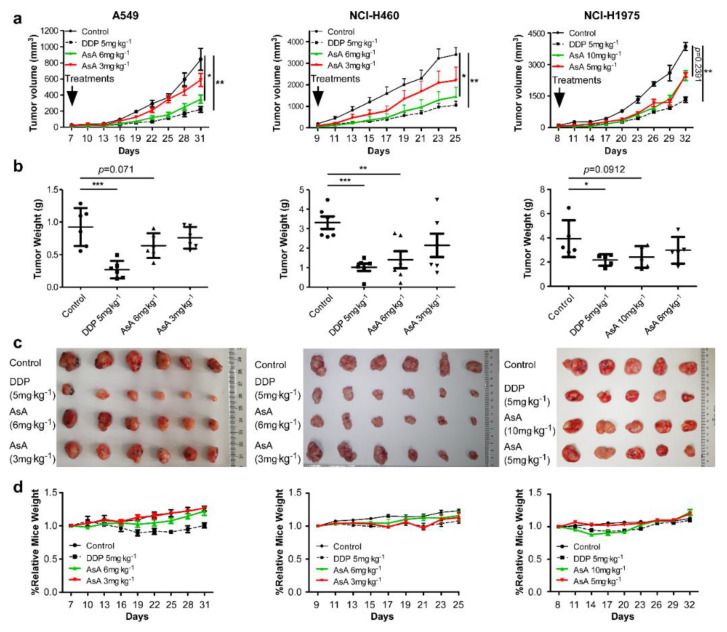
AsA suppressed lung cancer growth in the mouse xenograft model. (**a**) Tumor volumes of A549, NCI-H460 and NCI-H1975 xenografts following treatment with AsA (3 mg·kg^−1^, 6 mg·kg^−1^, or 5 mg·kg^−1^, 10 mg·kg^−1^), DDP (5 mg·kg^−1^) once every three days. The average volume and standard deviation are plotted, the statistical comparison vs. vehicle-treated control is shown by t-test. * *p* < 0.05, ** *p* < 0.01, *** *p* < 0.001. (**b**) The tumor weight of each group of mice. * *p* < 0.05, ** *p* < 0.01, *** *p* < 0.001. (**c**) Subcutaneous tumors formed by the indicated cells were dissected and imaged. (**d**) The body weight change curves of each group of mice.

**Figure 3 marinedrugs-18-00494-f003:**
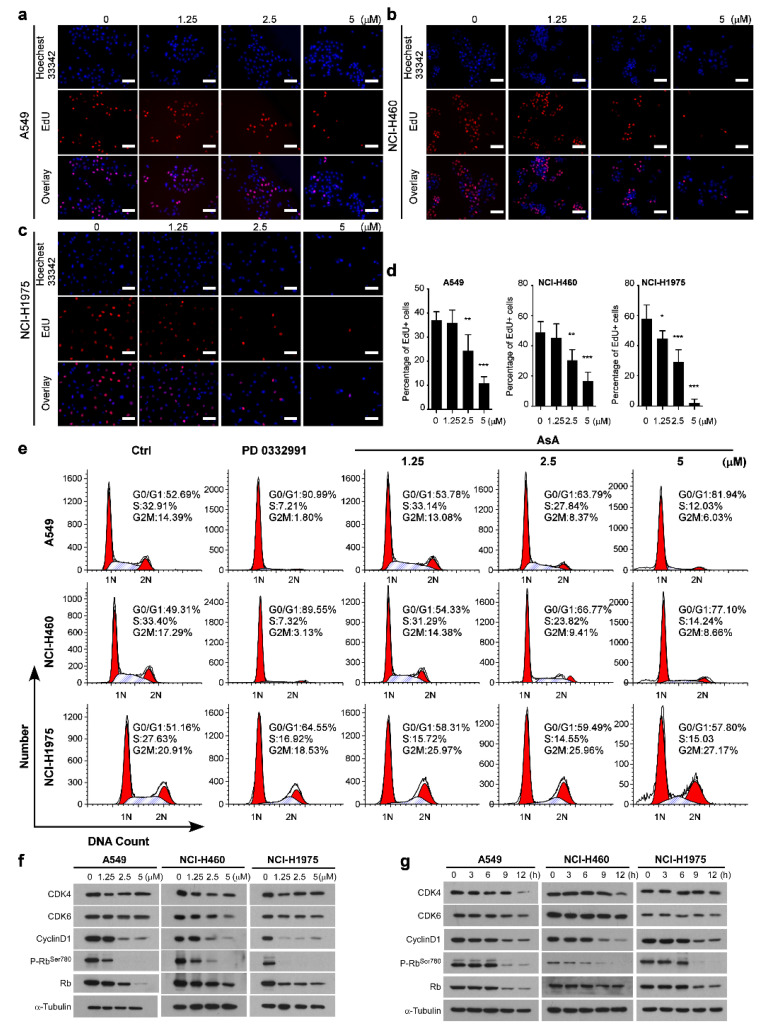
AsA induces G1 phase cell cycle arrest in human lung cancer cells. (**a–d**) The lung cancer cells’ proliferation capacity detected by 5-ethynyl-20-deoxyuridine (EdU) assay. A549, NCI-H460 and NCI-H1975 cells were treated by AsA at indicated concentrations for 12 h, * *p* < 0.05, ** *p* < 0.01, *** *p* < 0.001. EdU assay data are mean ± SD and representative of ≥ 3 experimental replicates. (**e**) A549, NCI-H460 and NCI-H1975 cells were treated with AsA (0, 1.25, 2.5 and 5 μM) or PD 0,332,991 (a CDK4/6 inhibitor as a positive control, 0.05 μM) for 12 h and subjected to DNA content analysis using a FACSCAN flow cytometer. (**f**) Proteins involved in G1/S phase transition were analyzed by Western blotting. Cells were treated with AsA (0, 1.25, 2.5, 5 μM) for 12 h. Equal protein loading was evaluated by α-Tubulin. (**g**) A549, NCI-H460 and NCI-H1975 cells were treated with AsA (5μM) for different times (0, 3, 6, 9, 12 h). Proteins involved in G1/S phase transition were analyzed by Western blotting. Equal protein loading was evaluated by α-Tubulin.

**Figure 4 marinedrugs-18-00494-f004:**
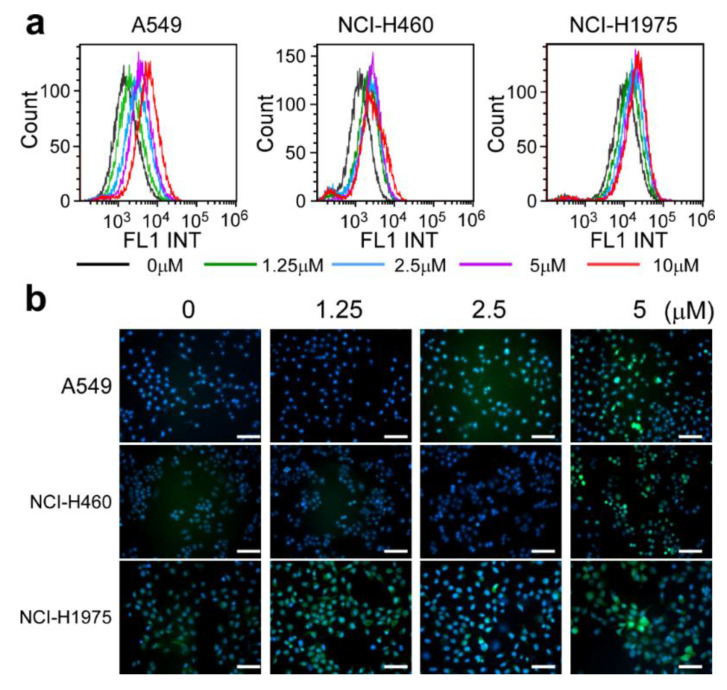
AsA induces reactive oxygen species (ROS) generation in human lung cancer cells. (**a**) Lung cancer cells were treated with different doses of AsA (0, 1.25, 2.5, 5, 10 μM) for 2 h. The mean fluorescent intensity of 2′,7′-dichlorodihydrofluorescein diacetate (DCFH-DA) staining is shown in A549, NCI-H460 and NCI-H1975 cells. ROS were detected by flow cytometry. (**b**) Lung cancer cells were treated with different doses of AsA (0, 1.25, 2.5, 5 μM) for 2 h. ROS were detected by fluorescence microscopy (magnification, 100×).

**Figure 5 marinedrugs-18-00494-f005:**
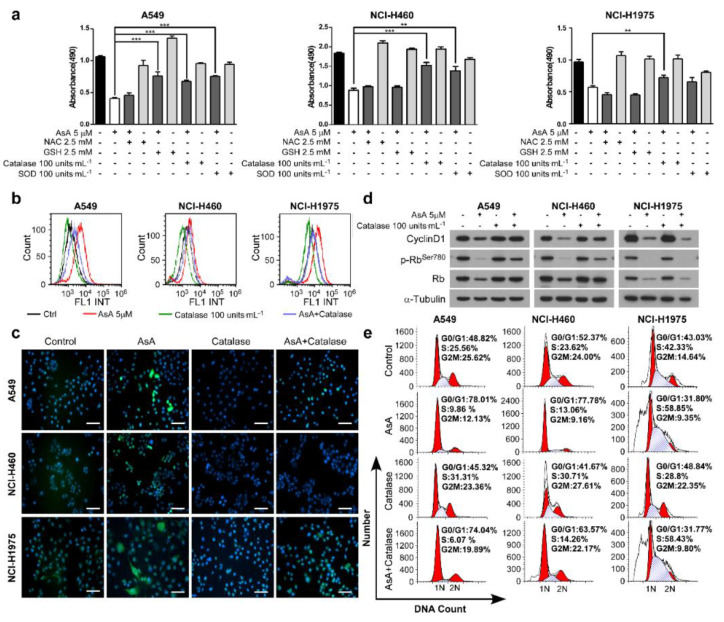
The ROS scavengers diminish AsA-induced ROS generation, and rescue growth inhibition and G1/S arrest in AsA-treated lung cancer cells. (**a**) The inhibition of cell growth induced by AsA was diminished by ROS scavengers, such as glutathione (GSH), catalase and superoxide dismutase (SOD), except for N-acetyl-L-cysteine (NAC). After the pretreatment of indicated ROS scavengers for 2 h, respectively, further treatments as indicated were given to the cells (8 × 10^3^ per well) for 24 h. The absorbance was measured as described in Methods and compared with the untreated control cells. Data are shown as mean ± SD and the asterisks (**, ***) indicate a significant (*p* < 0.01, *p* < 0.001, respectively) decrease in the viability of treated cells compared with the untreated control cells. (**b** and **c**) A549, NCI-H460 and NCI-H1975 cells were treated with AsA (5 μM) and catalase (100 units·mL^−1^) each alone or in combination for 2 h. ROS were detected by flow cytometry (**b**) or fluorescence microscopy (**c**, magnification, 100×). (**d**) Western blot analysis was performed in cells pretreated with catalase and co-treatment with AsA for 12 h. (**e**) A549, NCI-H460 and NCI-H1975 cells were treated with catalase (100 units·mL^−1^) and AsA (5μM) each alone or in combination for 12 h and subjected to DNA content analysis using a FACSCAN flow cytometer.

**Figure 6 marinedrugs-18-00494-f006:**
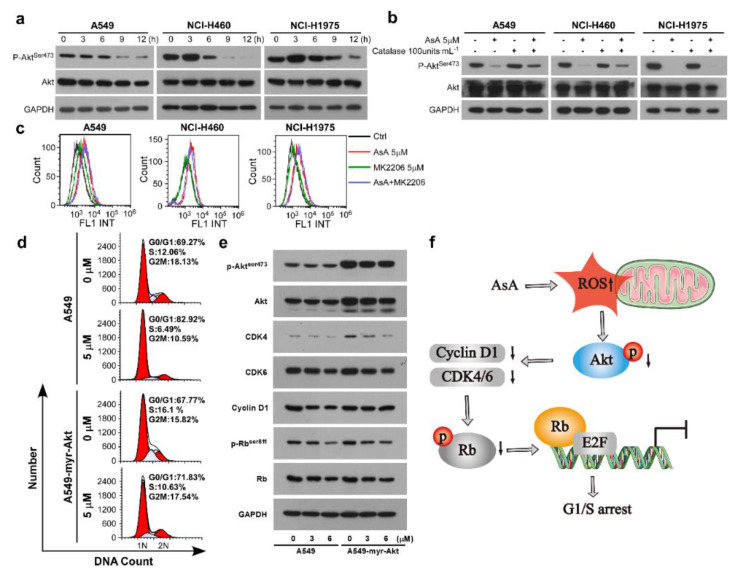
AsA induces the ROS-dependent inactivation of PI3K/Akt pathway in lung cancer cells. (**a**) A549, NCI-H460 and NCI-H1975 cells were treated with AsA (5 μM) for different times (0, 3, 6, 9, 12 h). P-Akt (Ser473) and T-Akt were analyzed by Western blotting. Equal protein loading was evaluated by α-Tubulin. (**b**) Western blot analysis was performed in A549, NCI-H460 and NCI-H1975 cells pretreated with AsA and catalase each alone or in combination for 12 h. (**c**) A549, NCI-H460 and NCI-H1975 cells were treated with AsA and MK2206 each alone or in combination for 2 h. ROS were detected by flow cytometry. (**d**) A549 and A549-myr-Akt cells were treated with or without AsA for 12 h and subjected to DNA content analysis using a FACSCAN flow cytometer. (**e**) Western blot analysis was performed in 549 and A549-myr-Akt cells pretreated with AsA (0, 3, 6 μM) for 12 h. (**f**) A schematic diagram of the proposed mechanisms of AsA-induced cell cycle G1 phase arrest in lung cancer cells.

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
