# Peer review of "A Marine Alkaloid, Ascomylactam A, Suppresses Lung Tumorigenesis via Inducing Cell Cycle G1/S Arrest through ROS/Akt/Rb Pathway"

_marinedrugs, 2020, doi:10.3390/md18100494_

Round 1

Reviewer 1 Report

The research of Wang et al. reports on the anticancer in vitro and in vivo activity of ascomylactams A in the lung cancer cell models. The authors have explored the importance of ROS/Akt/Rb pathway for the ability of the compound to induce a cell cycle arrest of the cancer cells.

Major points:

  • Introduction lacks sufficient information on the compound background and on what is already known on its activity and mode of action.
  • Allover the manuscript the data should be represented as mean +/- SD. This issue especially refers to the FACS cell cycle data.
  • The apoptotic markers such as cleaved caspase(s), PARP etc. should be accessed as well. AsA is most probably also affects cell viability – not only cell proliferation! The observed downregulation of the proteins may be an unspecific result of the cell death-related events. This issue should be addressed and the cell death markers should be examined along with the proteins of interest.
  • The Discussion is too superficial and is basically a repetition of the Results section. In vivo results, side effects (if any), as well as the correlation of in vivo and in vitro results are not discussed. This section should be extensively improved.

Minor points:

  • The abbreviations (e.g. NAC, SOD, etc.) should be explained in the manuscript text when they are mentioned for the first time.
  • Figure 3 – what is PD0332991?
  • Please quantify data from figures 4a, 5b, 6c.
  • Please explain how the ROS was detected on Fig. 4b
  • English style should be improved overall. Several examples: Line 160-161: should be “except for”. Lines 196-200: should be double-checked and rephrased; Lines 219-226: the style should be improved; Line 250: should be “activity” instead of "inactivity", etc.

Reviewer 2 Report

This manuscript is a well written report on ascomylactam A isolated from the mangrove endophytic fungus Didymella sp. Ascomylactam A was reported by the same group of this manuscript as a new 13-membered-ring macrocyclic alkaloid isolated from the mangrove endophytic fungus Didymella sp. CYSK-4. This well written paper, while not ground breaking in its findings or methodologies, is certainly worthy of publication with minor revision.

Minor comments:
Page 1 and in whole manuscript: Ascomylactams A should be written as ascomylactam A.

Page 1, lines 23 and 60 and other pages: Authors insist that ascomylactam A is a novel or a new compound but it was already reported in 2019 by the same authors of this paper. So it is not a novel or a new compound anymore. The sentences should be rewritten as ascomylactam A was reported for the first time as a new 13-membered-ring macrocyclic alkaloid in 2019 from ~~~[13].
Page 1 line 67: phrase à phase

Page 3, Fig 1b: cell line à cell lines

Page 3, line 85: represent à represents

Page 4, line 99: DDP (5mg·kg-1) à DDP (cisplatin, 5mg·kg-1)

Page 4, line 112: DDP (cisplatin, 5mg·kg-1) à DDP (5mg·kg-1)

Page 4, line 122~123: the analysis of the cell cycle distribution by flow cytometry showed that, with the treatment of AsA for 12 h, A549 cells resulted in an increased percentage of cell population in G0/G1 phase ~~ à the analysis of the cell cycle distribution by flow cytometry, with the treatment of AsA for 12 h against A549 cells, resulted in an increased percentage of cell population in G0/G1 phase ~~.

Page 8, line 227: ROS play à ROS plays

In references, authors should follow the journal rules.

Reviewer 3 Report

Through the research described in this manuscript, the mode of action of ascomylactam against lung cancers is revealed. The experiments were designed rationally and performed properly, and the results are depicted efficiently. The conclusion that ascomlylatam A induces cell cycle arrest through the inactivation of PI3K/Akt pathway caused by the induction of ROS is based upon sound experimental evidences. This manuscript is qualified enough to be published, however, the overall proofreading is recommended. The points listed below should be considered during the proofreading

  1. L2, L23, L36, L60, L83 ; ‘Ascomylactams A’ should be corrected as ‘Ascomylactam A’ (singular, not plural)
  2. L54-56 ; Correction of a sentence might be needed, especially the phrase ‘than do non-malignant cells’
  3. L71; ‘the chemical structure shown’ in parentheses should be omitted.
  4. L80, L224 ; The sentences starting with ‘And’ don’t look good.
  5. Figure 1b; The IC50 values are not readable due to the small text size.
  6. Figrures 3-6 ; The use of capital letters in the title should be reexamined.

Reviewer 4 Report

This paper describes a detailed bioactivity results of the most potent Ascomylactam A among the Ascomylactams whose isolated compounds have been determined 13-membered-ring macrocyclic alkaloids by She’s group (ref. 13). It has been demonstrated that Ascomylactam A effectively inhibits the cell growth of lung cancer cells. Furthermore, it has also been shown to induce G1/S arrest, and modulate the inactivity of the Akt/Cyclin D1 pathway in lung cancer cells through a ROS-dependent mechanism. I think that many researchers are interested in this result, and this work is valuable for publication in this journal. This reviewer also recommends considering the minor things noted below:

Should be corrected “Ascomylactams A” to “Ascomylactam A” in text.
